# Dynamics of stochastic gradient descent for two-layer neural networks in the teacher-student setup

**Sebastian Goldt[1], Madhu S. Advani[2], Andrew M. Saxe[3]**
**Florent Krzakala[4], Lenka Zdeborová[1]**
[1] Institut de Physique Théorique, CNRS, CEA, Université Paris-Saclay, Saclay, France
[2] Center for Brain Science, Harvard University, Cambridge, MA 02138, USA
[3] Department of Experimental Psychology, University of Oxford, Oxford, United Kingdom
[4] Laboratoire de Physique Statistique, Sorbonne Universités,
Université Pierre et Marie Curie Paris 6, Ecole Normale Supérieure, 75005 Paris, France

## Abstract

Deep neural networks achieve stellar generalisation even when they have enough parameters to easily fit all their training data. We study this phenomenon by analysing the dynamics and the performance of over-parameterised two-layer neural networks in the teacher-student setup, where one network, the student, is trained on data generated by another network, called the teacher. We show how the dynamics of stochastic gradient descent (SGD) is captured by a set of differential equations and prove that this description is asymptotically exact in the limit of large inputs. Using this framework, we calculate the final generalisation error of student networks that have more parameters than their teachers. We find that the final generalisation error of the student increases with network size when training only the first layer, but stays constant or even decreases with size when training both layers. We show that these different behaviours have their root in the different solutions SGD finds for different activation functions. Our results indicate that achieving good generalisation in neural networks goes beyond the properties of SGD alone and depends on the interplay of at least the algorithm, the model architecture, and the data set.

Deep neural networks behind state-of-the-art results in image classification and other domains have one thing in common: their size. In many applications, the free parameters of these models outnumber the samples in their training set by up to two orders of magnitude[1,2]. Statistical learning theory suggests that such heavily over-parameterised networks generalise poorly without further regularisation[3–9], yet empirical studies consistently find that increasing the size of networks to the point where they can easily fit their training data and beyond does not impede their ability to generalise well, even without any explicit regularisation[10–12]. Resolving this paradox is arguably one of the big challenges in the theory of deep learning.

One tentative explanation for the success of large networks has focused on the properties of stochastic gradient descent (SGD), the algorithm routinely used to train these networks. In particular, it has been proposed that SGD has an implicit regularisation mechanism that ensures that solutions found by SGD generalise well irrespective of the number of parameters involved, for models as diverse as (over-parameterised) neural networks[10,13], logistic regression[14] and matrix factorisation models[15,16].

In this paper, we analyse the dynamics of one-pass (or online) SGD in two-layer neural networks. We focus in particular on the influence of over-parameterisation on the final generalisation error. We use the teacher-student framework[17,18], where a training data set is generated by feeding random inputs through a two-layer neural network with $M$ hidden units called the *teacher*. Another neural network, the *student*, is then trained using SGD on that data set. The generalisation error is defined as the mean

squared error between teacher and student outputs, averaged over all of input space. We will focus on student networks that have a larger number of hidden units $K \geq M$ than their teacher. This means that the student can express much more complex functions than the teacher function they have to learn; the students are thus over-parameterised with respect to the generative model of the training data in a way that is simple to quantify. We find this definition of over-parameterisation cleaner in our setting than the oft-used comparison of the number of parameters in the model with the number of samples in the training set, which is not well justified for non-linear functions. Furthermore, these two numbers surely cannot fully capture the complexity of the function learned in practical applications.

The teacher-student framework is also interesting in the wake of the need to understand the effectiveness of neural networks and the limitations of the classical approaches to generalisation[11]. Traditional approaches to learning and generalisation are data agnostic and seek worst-case type bounds[19]. On the other hand, there has been a considerable body of theoretical work calculating the generalisation ability of neural networks for data arising from a probabilistic model, particularly within the framework of statistical mechanics[17,18,20–22]. Revisiting and extending the results that have emerged from this perspective is currently experiencing a surge of interest[23–28].

In this work we consider two-layer networks with a large input layer and a finite, but arbitrary, number of hidden neurons. Other limits of two-layer neural networks have received a lot of attention recently. A series of papers[29–32] studied the mean-field limit of two-layer networks, where the number of neurons in the hidden layer is very large, and proved various general properties of SGD based on a description in terms of a limiting partial differential equation. Another set of works, operating in a different limit, have shown that infinitely wide over-parameterised neural networks trained with gradient-based methods effectively solve a kernel regression[33–38], without any feature learning. Both the mean-field and the kernel regime crucially rely on having an infinite number of nodes in the hidden layer, and the performance of the networks strongly depends on the detailed scaling used[38,39]. Furthermore, a very wide hidden layer makes it hard to have a student that is larger than the teacher in a quantifiable way. This leads us to consider the opposite limit of large input dimension and finite number of hidden units.

Our **main contributions** are as follows:

*(i)* The dynamics of SGD (online) learning by two-layer neural networks in the teacher-student setup was studied in a series of classic papers[40–44] from the statistical physics community, leading to a heuristic derivation of a set of coupled ordinary differential equations (ODE) that describe the *typical* time-evolution of the generalisation error. *We provide a rigorous foundation of the ODE approach to analysing the generalisation dynamics in the limit of large input size by proving their correctness.*

*(ii)* These works focused on training only the first layer, mainly in the case where the teacher network has the same number of hidden units and the student network, $K = M$. *We generalise their analysis to the case where the student's expressivity is considerably larger than that of the teacher* in order to investigate the *over-parameterised regime $K > M$.*

*(iii) We provide a detailed analysis of the dynamics of learning and of the generalisation when only the first layer is trained.* We derive a reduced set of coupled ODE that describes the generalisation dynamics for any $K \geq M$ and obtain analytical expressions for the asymptotic generalisation error of networks with linear and sigmoidal activation functions. Crucially, we find that with all other parameters equal, the final generalisation error *increases* with the size of the student network. In this case, SGD alone thus does not seem to be enough to regularise larger student networks.

*(iv) We finally analyse the dynamics when learning both layers.* We give an analytical expression for the final generalisation error of sigmoidal networks and find evidence that suggests that SGD finds solutions which amount to performing an effective model average, thus improving the generalisation error upon over-parameterisation. In linear and ReLU networks, we experimentally find that the generalisation error does change as a function of $K$ when training both layers. However, there exist student networks with better performance that are fixed points of the SGD dynamics, but are not reached when starting SGD from initial conditions with small, random weights.

Crucially, we find this range of different behaviours while keeping the training algorithm (SGD) the same, changing only the activation functions of the networks and the parts of the network that are trained. Our results clearly indicate that the implicit regularisation of neural networks in our setting goes beyond the properties of SGD alone. Instead, a full understanding of the generalisation properties of even very simple neural networks requires taking into account the interplay of at least

the algorithm, the network architecture, and the data set used for training, setting up a formidable research programme for the future.

**Reproducibility —** We have packaged the implementation of our experiments and our ODE integrator into a user-friendly library with example programs at `https://github.com/sgoldt/nn2pp`. All plots were generated with these programs, and we give the necessary parameter values beneath each plot.

# 1   Online learning in teacher-student neural networks

We consider a supervised regression problem with training set $\mathcal{D} = \{(x^\mu, y^\mu)\}$ with $\mu = 1, \ldots, P$. The components of the inputs $x^\mu \in \mathbb{R}^N$ are i.i.d. draws from the standard normal distribution $\mathcal{N}(0,1)$. The scalar labels $y^\mu$ are given by the output of a network with $M$ hidden units, a non-linear activation function $g : \mathbb{R} \to \mathbb{R}$ and fixed weights $\theta^* = (v^* \in \mathbb{R}^M, w^* \in \mathbb{R}^{M \times N})$ with an additive output noise $\zeta^\mu \sim \mathcal{N}(0,1)$, called the *teacher* (see also Fig. 1a):

$$y^\mu \equiv \phi(x^\mu, \theta^*) + \sigma\zeta^\mu, \qquad \text{where} \quad \phi(x, \theta^*) = \sum_{m=1}^{M} v_m^* g\left(\frac{w_m^* x}{\sqrt{N}}\right) = \sum_m v_m^* g(\rho_m), \quad (1)$$

where $w_m^*$ is the $m$th row of $w^*$, and the local field of the $m$th teacher node is $\rho_m \equiv w_m^* x / \sqrt{N}$. We will analyse three different network types: sigmoidal with $g(x) = \operatorname{erf}(x/\sqrt{2})$, ReLU with $g(x) = \max(x, 0)$, and linear networks where $g(x) = x$.

A second two-layer network with $K$ hidden units and weights $\theta = (v \in \mathbb{R}^K, w \in \mathbb{R}^{K \times N})$, called the *student*, is then trained using SGD on the quadratic training loss $E(\theta) \propto \sum_{\mu=1}^{P} [\phi(x^\mu, \theta) - y^\mu]^2$. We emphasise that the student network may have a larger number of hidden units $K \geq M$ than the teacher and thus be over-parameterised with respect to the generative model of its training data.

The SGD algorithm defines a Markov process $X^\mu \equiv [v^*, w^*, v^\mu, w^\mu]$ with update rule given by the coupled SGD recursion relations

$$w_k^{\mu+1} = w_k^\mu - \frac{\eta_w}{\sqrt{N}} v_k^\mu g'(\lambda_k^\mu) \Delta^\mu x^\mu, \qquad (2)$$

$$v_k^{\mu+1} = v_k^\mu - \frac{\eta_v}{N} g(\lambda_k^\mu) \Delta^\mu. \qquad (3)$$

We can choose different learning rates $\eta_v$ and $\eta_w$ for the two layers and denote by $g'(\lambda_k^\mu)$ the derivative of the activation function evaluated at the local field of the student's $k$th hidden unit $\lambda_k^\mu \equiv w_k x^\mu / \sqrt{N}$, and we defined the error term $\Delta^\mu \equiv \sum_k v_k^\mu g(\lambda_k^\mu) - \sum_m v_m^* g(\rho_m^\mu) - \sigma\zeta^\mu$. We will use the indices $i, j, k, \ldots$ to refer to student nodes, and $n, m, \ldots$ to denote teacher nodes. We take initial weights at random from $\mathcal{N}(0,1)$ for sigmoidal networks, while initial weights have variance $1/\sqrt{N}$ for ReLU and linear networks.

The key quantity in our approach is the *generalisation error* of the student with respect to the teacher:

$$\epsilon_g(\theta, \theta^*) \equiv \frac{1}{2} \left\langle [\phi(x, \theta) - \phi(x, \theta^*)]^2 \right\rangle, \qquad (4)$$

where the angled brackets $\langle \cdot \rangle$ denote an average over the input distribution. We can make progress by realising that $\epsilon_g(\theta^*, \theta)$ can be expressed as a function of a set of macroscopic variables, called *order parameters* in statistical physics,[21,40,41]

$$Q_{ik}^\mu \equiv \frac{w_i^\mu w_k^\mu}{N}, \quad R_{in}^\mu \equiv \frac{w_i^\mu w_n^*}{N} \quad \text{and} \quad T_{nm} \equiv \frac{w_n^* w_m^*}{N}, \qquad (5)$$

together with the second-layer weights $v^*$ and $v^\mu$. Intuitively, the teacher-student overlaps $R^\mu = [R_{in}^\mu]$ measure the similarity between the weights of the $i$th student node and the $n$th teacher node. The matrix $Q_{ik}$ quantifies the overlap of the weights of different student nodes with each other, and the corresponding overlap of the teacher nodes are collected in the matrix $T_{nm}$. We will find it convenient to collect all order parameters in a single vector

$$m^\mu \equiv (R^\mu, Q^\mu, T, v^*, v^\mu), \qquad (6)$$

and we write the full expression for $\epsilon_g(m^\mu)$ in the SM, Eq. (S31).

In a series of classic papers, Biehl, Schwarze, Saad, Solla and Riegler[40–44] derived a closed set of ordinary differential equations for the time evolution of the order parameters $m$ (see SM Sec. B). Together with the expression for the generalisation error $\epsilon_g(m^\mu)$, these equations give a complete description of the generalisation dynamics of the student, which they analysed for the special case $K = M$ when only the first layer is trained[42,44]. Our first contribution is to provide a rigorous foundation for these results under the following assumptions:

**(A1)** Both the sequences $x^\mu$ and $\zeta^\mu$, $\mu = 1, 2, \ldots$, are i.i.d. random variables; $x^\mu$ is drawn from a normal distribution with mean 0 and covariance matrix $\mathbb{I}_N$, while $\zeta^\mu$ is a Gaussian random variable with mean zero and unity variance;

**(A2)** The function $g(x)$ is bounded and its derivatives up to and including the second order exist and are bounded, too;

**(A3)** The initial macroscopic state $m^0$ is deterministic and bounded by a constant;

**(A4)** The constants $\sigma$, $K$, $M$, $\eta_w$ and $\eta_v$ are all finite.

The correctness of the ODE description is then established by the following theorem:

**Theorem 1.1.** *Choose $T > 0$ and define $\alpha \equiv \mu/N$. Under assumptions (A1) – (A4), and for any $\alpha > 0$, the macroscopic state $m^\mu$ satisfies*

$$\max_{0 \leq \mu \leq NT} \mathbb{E} \ ||m^\mu - m(\alpha)|| \leq \frac{C(T)}{\sqrt{N}} \,, \tag{7}$$

*where $C(T)$ is a constant depending on $T$, but not on $N$, and $m(\alpha)$ is the unique solution of the ODE*

$$\frac{\mathrm{d}}{\mathrm{d}t} m(\alpha) = f(m(\alpha)) \tag{8}$$

*with initial condition $m^*$. In particular, we have*

$$\frac{\mathrm{d}R_{in}}{\mathrm{d}\alpha} \equiv f_R(m(\alpha)) = \eta v_i \langle \Delta g'(\lambda_i)\rho_n \rangle \,, \tag{9a}$$

$$\frac{\mathrm{d}Q_{ik}}{\mathrm{d}\alpha} \equiv f_Q(m(\alpha)) = \eta v_i \langle \Delta g'(\lambda_i)\lambda_k \rangle + \eta v_k \langle \Delta g'(\lambda_k)\lambda_i \rangle$$
$$+ \eta^2 v_i v_k \langle \Delta^2 g'(\lambda_i)g'(\lambda_k) \rangle + \eta^2 v_i v_k \sigma^2 \langle g'(\lambda_i)g'(\lambda_k) \rangle \,, \tag{9b}$$

$$\frac{\mathrm{d}v_i}{\mathrm{d}\alpha} \equiv f_v(m(\alpha)) = \eta_v \langle \Delta g(\lambda_i) \rangle. \tag{9c}$$

*where all $f(m(\alpha))$ are uniformly Lipschitz continuous in $m(\alpha)$. We are able to close the equations because we can express averages in Eq. (9) in terms of only $m(\alpha)$.*

We prove Theorem 1.1 using the theory of convergence of stochastic processes and a coupling trick introduced recently by Wang et al.[45] in Sec. A of the SM. The content of the theorem is illustrated in Fig. 1b, where we plot $\epsilon_g(\alpha)$ obtained by numerically integrating (9) (solid) and from a single run of SGD (2) (crosses) for sigmoidal students and varying $K$, which are in very good agreement.

Given a set of non-linear, coupled ODE such as Eqns. (9), finding the asymptotic fixed points analytically to compute the generalisation error would seem to be impossible. In the following, we will therefore focus on analysing the asymptotic fixed points found by numerically integrating the equations of motion. The form of these fixed points will reveal a drastically different dependence of the test error on the over-parameterisation of neural networks with different activation functions in the different setups we consider, despite them all being trained by SGD. This highlights the fact that good generalisation goes beyond the properties of *just* the algorithm. Second, knowledge of these fixed points allows us to make analytical and quantitative predictions for the asymptotic performance of the networks which agree well with experiments. We also note that several recent theorems[29–31] about the global convergence of SGD do not apply in our setting because we have a finite number of hidden units.

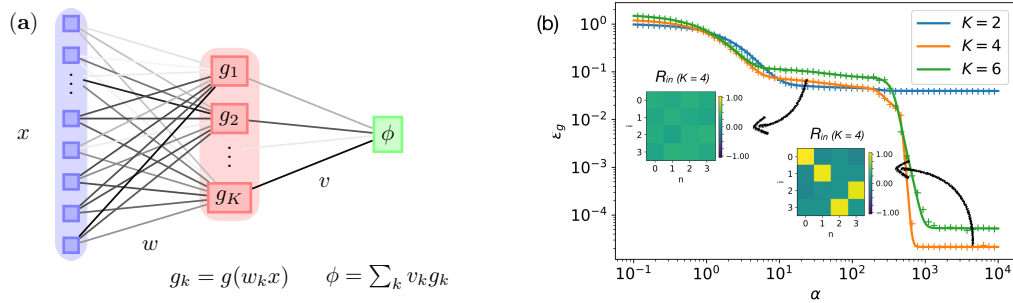

$$g_k = g(w_k x) \qquad \phi = \sum_k v_k g_k$$

Figure 1: **The analytical description of the generalisation dynamics of sigmoidal networks matches experiments. (a)** We consider two-layer neural networks with a very large input layer. **(b)** We plot the learning dynamics $\epsilon_g(\alpha)$ obtained by integration of the ODEs (9) (solid) and from a single run of SGD (2) (crosses) for students with different numbers of hidden units $K$. The insets show the values of the teacher-student overlaps $R_{in}$ (5) for a student with $K = 4$ at the two times indicated by the arrows. $N = 784, M = 4, \eta = 0.2$.

## 2 Asymptotic generalisation error of Soft Committee machines

We will first study networks where the second layer weights are fixed at $v_m^* = v_k = 1$. These networks are called a *Soft Committee Machine* (SCM) in the statistical physics literature [18,27,40–42,44]. One notable feature of $\epsilon_g(\alpha)$ in SCMs is the existence of a long plateau with sub-optimal generalisation error during training. During this period, all student nodes have roughly the same overlap with all the teacher nodes, $R_{in} = \text{const.}$ (left inset in Fig. 1b). As training continues, the student nodes "specialise" and each of them becomes strongly correlated with a single teacher node (right inset), leading to a sharp decrease in $\epsilon_g$. This effect is well-known for both batch and online learning [18] and will be key for our analysis.

Let us now use the equations of motion (9) to analyse the asymptotic generalisation error of neural networks $\epsilon_g^*$ after training has converged and in particular its scaling with $L = K - M$. Our first contribution is to reduce the remaining $K(K + M)$ equations of motion to a set of eight coupled differential equations for any combination of $K$ and $M$ in Sec. C. This enables us to obtain a closed-form expression for $\epsilon_g^*$ as follows.

In the absence of output noise ($\sigma = 0$), the generalisation error of a student with $K \geq M$ will asymptotically tend to zero as $\alpha \to \infty$. On the level of the order parameters, this corresponds to reaching a stable fixed point of (9) with $\epsilon_g = 0$. In the presence of small output noise $\sigma > 0$, this fixed point becomes unstable and the order parameters instead converge to another, nearby fixed point $m^*$ with $\epsilon_g(m^*) > 0$. The values of the order parameters at that fixed point can be obtained by perturbing Eqns. (9) to first order in $\sigma$, and the corresponding generalisation error $\epsilon_g(m^*)$ turns out to be in excellent agreement with the generalisation error obtained when training a neural network using (2) from random initial conditions, which we show in Fig. 2a.

**Sigmoidal networks.** We have performed this calculation for teacher and student networks with $g(x) = \text{erf}(x/\sqrt{2})$. We relegate the details to Sec. C.2, and content us here to state the asymptotic value of the generalisation error to first order in $\sigma^2$,

$$\epsilon_g^* = \frac{\sigma^2 \eta}{2\pi} f(M, L, \eta) + \mathcal{O}(\sigma^3), \tag{10}$$

where $f(M, L, \eta)$ is a lengthy rational function of its variables. We plot our result in Fig. 2a together with the final generalisation error obtained in a single run of SGD (2) for a neural network with initial weights drawn i.i.d. from $\mathcal{N}(0, 1)$ and find excellent agreement, which we confirmed for a range of values for $\eta$, $\sigma$, and $L$.

One notable feature of Fig. 2a is that with all else being equal, SGD alone fails to regularise the student networks of increasing size in our setup, instead yielding students whose generalisation error increases linearly with $L$. One might be tempted to mitigate this effect by simultaneously decreasing the learning rate $\eta$ for larger students. However, lowering the learning rate incurs longer training

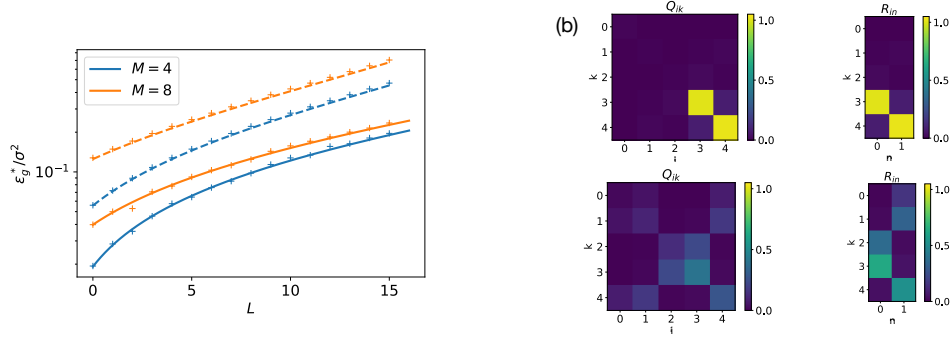

Figure 2: **The asymptotic generalisation error of Soft Committee Machines increases with the network size.** $N = 784, \eta = 0.05, \sigma = 0.01$. **(a)** Our theoretical prediction for $\epsilon_g^*/\sigma^2$ for sigmoidal (solid) and linear (dashed), Eqns. (10) and (12), agree perfectly with the result obtained from a single run of SGD (2) starting from random initial weights (crosses). **(b)** The final overlap matrices $Q$ and $R$ (5) at the end of an experiment with $M = 2, K = 5$. Networks with sigmoidal activation function (top) show clear signs of specialisation as described in Sec. 2. ReLU networks (bottom) instead converge to solutions where all of the student's nodes have finite overlap with teacher nodes.

times, which requires more data for online learning. This trade-off is also found in statistical learning theory, where models with more parameters (higher $L$) and thus a higher complexity class (*e.g.* VC dimension or Rademacher complexity[4]) generalise just as well as smaller ones when given more data. In practice, however, more data might not be readily available, and we show in Fig. S2 of the SM that even when choosing $\eta = 1/K$, the generalisation error still increases with $L$ before plateauing at a constant value.

We can gain some intuition for the scaling of $\epsilon_g^*$ by considering the asymptotic overlap matrices $Q$ and $R$ shown in the left half of Fig. 2b. In the over-parameterised case, $L = K - M$ student nodes are effectively trying to specialise to teacher nodes which do not exist, or equivalently, have weights zero. These $L$ student nodes do not carry any information about the teachers output, but they pick up fluctuations from output noise and thus increase $\epsilon_g^*$. This intuition is borne out by an expansion of $\epsilon_g^*$ in the limit of small learning rate $\eta$, which yields

$$\epsilon_g^* = \frac{\sigma^2 \eta}{2\pi} \left( L + \frac{M}{\sqrt{3}} \right) + \mathcal{O}(\eta^2), \tag{11}$$

which is indeed the sum of the error of $M$ independent hidden units that are specialised to a single teacher hidden unit, and $L = K - M$ superfluous units contributing each the error of a hidden unit that is "learning" from a hidden unit with zero weights $w_m^* = 0$ (see also Sec. D of the SM).

**Linear networks.** Two possible explanations for the scaling $\epsilon_g^* \sim L$ in sigmoidal networks may be the specialisation of the hidden units or the fact that teacher and student network can implement functions of different range if $K \neq M$. To test these hypotheses, we calculated $\epsilon_g^*$ for linear neural networks[46,47] with $g(x) = x$. Linear networks lack a specialisation transition[27] and their output range is set by the magnitude of their weights, rather than their number of hidden units. Following the same steps as before, a perturbative calculation in the limit of small noise variance $\sigma^2$ yields

$$\epsilon_g^* = \frac{\eta \sigma^2 (L + M)}{4 - 2\eta(L + M)} + \mathcal{O}(\sigma^3). \tag{12}$$

This result is again in perfect agreement with experiments, as we demonstrate in Fig. 2a. In the limit of small learning rates $\eta$, Eq. (10) simplifies to yield the same scaling as for sigmoidal networks,

$$\epsilon_g^* = \frac{1}{4}\eta \sigma^2 (L + M) + \mathcal{O}\left(\eta^2\right). \tag{13}$$

This shows that the scaling $\epsilon_g^* \sim L$ is not just a consequence of either specialisation or the mismatched range of the networks' output functions. The optimal number of hidden units for linear networks is $K = 1$ for all $M$, because linear networks implement an effective linear transformation with an

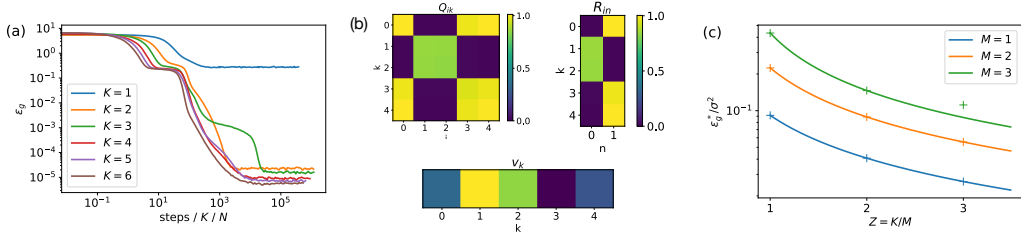

Figure 3: **The performance of sigmoidal networks improves with network size when training both layers with SGD. (a)** Generalisation dynamics observed experimentally for students with increasing $K$, with all other parameters being equal. ($N = 500, M = 2, \eta = 0.05, \sigma = 0.01, v^* = 4$). **(b)** Overlap matrices $Q$, $R$, and second layer weights $v_k$ of the student at the end of the run with $K = 5$ shown in (a). **(c)** Theoretical prediction for $\epsilon_g^*$ (solid) against $\epsilon_g^*$ observed after integration of the ODE until convergence (crosses) (9) ($\sigma = 0.01, \eta = 0.2, v^* = 2$).

effective matrix $W = \sum_k w_k$. Adding hidden units to a linear network hence does not augment the class of functions it can implement, but it adds redundant parameters which pick up fluctuations from the teacher's output noise, increasing $\epsilon_g$.

**ReLU networks.** The analytical calculation of $\epsilon_g^*$, described above, for ReLU networks poses some additional technical challenges, so we resort to experiments to investigate this case. We found that the asymptotic generalisation error of a ReLU student learning from a ReLU teacher has the same scaling as the one we found analytically for networks with sigmoidal and linear activation functions: $\epsilon_g^* \sim \eta \sigma^2 L$ (see Fig. S3). Looking at the final overlap matrices $Q$ and $R$ for ReLU networks in the bottom half of Fig. 2b, we see that instead of the one-to-one specialisation of sigmoidal networks, all student nodes have a finite overlap with some teacher node. This is a consequence of the fact that it is much simpler to re-express the sum of $M$ ReLU units with $K \neq M$ ReLU units. However, there are still a lot of redundant degrees of freedom in the student, which all pick up fluctuations from the teacher's output noise and increase $\epsilon_g^*$.

**Discussion.** The key result of this section has been that the generalisation error of SCMs scales as

$$\epsilon_g^* \sim \eta \sigma^2 L. \tag{14}$$

Before moving on the full two-layer network, we discuss a number of experiments that we performed to check the robustness of this result (Details can be found in Sec. G of the SM). A standard regularisation method is adding weight decay to the SGD updates (2). However, we did not find a scenario in our experiments where weight decay improved the performance of a student with $L > 0$. We also made sure that our results persist when performing SGD with mini-batches. We investigated the impact of higher-order correlations in the inputs by replacing Gaussian inputs with MNIST images, with all other aspects of our setup the same, and the same $\epsilon_g$-$L$ curve as for Gaussian inputs. Finally, we analysed the impact of having a finite training set. The behaviour of linear networks and of non-linear networks with large but finite training sets did not change qualitatively. However, as we reduce the size of the training set, we found that the lowest asymptotic generalisation error was obtained with networks that have $K > M$.

## 3   Training both layers: Asymptotic generalisation error of a neural network

We now study the performance of two-layer neural networks when both layers are trained according to the SGD updates (2) and (3). We set all the teacher weights equal to a constant value, $v_m^* = v^*$, to ensure comparability between experiments. However, we train all $K$ second-layer weights of the student independently and do not rely on the fact that all second-layer teacher weights have the same value. Note that learning the second layer is not needed from the point of view of statistical learning: the networks from the previous section are already expressive enough to capture the students, and we are thus slightly increasing the over-parameterisation even further. Yet, we will see that the generalisation properties will be significantly enhanced.

**Sigmoidal networks.** We plot the generalisation dynamics of students with increasing $K$ trained on a teacher with $M = 2$ in Fig. 3a. Our first observation is that increasing the student size $K \geq M$ *decreases* the asymptotic generalisation error $\epsilon_g^*$, with all other parameters being equal, in stark contrast to the SCMs of the previous section.

A look at the order parameters after convergence in the experiments from Fig. 3a reveals the intriguing pattern of specialisation of the student's hidden units behind this behaviour, shown for $K = 5$ in Fig. 3b. First, note that all the hidden units of the student have non-negligible weights ($Q_{ii} > 0$). Two student nodes ($k = 1, 2$) have specialised to the first teacher node, *i.e.* their weights are very close to the weights of the first teacher node ($R_{10} \approx R_{20} \approx 0.85$). The corresponding second-layer weights approximately fulfil $v_1 + v_3 \approx v^*$. Summing the output of these two student hidden units is thus approximately equivalent to an empirical average of two estimates of the output of the teacher node. The remaining three student nodes all specialised to the second teacher node, and their outgoing weights approximately sum to $v^*$. This pattern suggests that SGD has found a set of weights for both layers where the student's output is a weighted average of several estimates of the output of the teacher's nodes. We call this the *denoising solution* and note that it resembles the solutions found in the mean-field limit of an infinite hidden layer[29,31] where the neurons become redundant and follow a distribution dynamics (in our case, a simple one with few peaks, as e.g. Fig. 1 in[31]).

We confirmed this intuition by using an ansatz for the order parameters that corresponds to a denoising solution to solve the equations of motion (9) perturbatively in the limit of small noise to calculate $\epsilon_g^*$ for sigmoidal networks after training both layers, similarly to the approach in Sec. 2. While this approach can be extended to any $K$ and $M$, we focused on the case where $K = ZM$ to obtain manageable expressions; see Sec. E of the SM for details on the derivation. While the final expression is again too long to be given here, we plot it with solid lines in Fig. 3c. The crosses in the same plot are the asymptotic generalisation error obtained by integration of the ODE (9) starting from random initial conditions, and show very good agreement.

While our result holds for any $M$, we note from Fig. 3c that the curves for different $M$ are qualitatively similar. We find a particular simple result for $M = 1$ in the limit of small learning rates, where:

$$\epsilon_g^* = \frac{\eta(\sigma v^*)^2}{2\sqrt{3}K\pi} + \mathcal{O}(\eta\sigma^2).$$  (15)

This result should be contrasted with the $\epsilon_g \sim K$ behaviour found for SCM.

Experimentally, we robustly observed that training both layers of the network yields better performance than training only the first layer with the second layer weights fixed to $v^*$. However, convergence to the denoising solution can be difficult for large students which might get stuck on a long plateau where their nodes are not evenly distributed among the teacher nodes. While it is easy to check that such a network has a higher value of $\epsilon_g$ than the denoising solution, the difference is small, and hence the driving force that pushes the student out of the corresponding plateaus is small, too. These observations demonstrate that in our setup, SGD does not always find the solution with the lowest generalisation error in finite time.

**ReLU and linear networks.** We found experimentally that $\epsilon_g^*$ remains constant with increasing $K$ in ReLU and in linear networks when training both layers. We plot a typical learning curve in green for linear networks in Fig. 4, but note that the figure shows qualitatively similar features for ReLU networks (Fig. S4). This behaviour was also observed in linear networks trained by *batch* gradient descent, starting from small initial weights[48]. While this scaling of $\epsilon_g^*$ with $K$ is an improvement over its increase with $K$ for the SCM, (blue curve), this is not the $1/K$ decay that we observed for sigmoidal networks. A possible explanation is the lack of specialisation in linear and ReLU networks (see Sec. 2), without which the denoising solution found in sigmoidal networks is not possible. We also considered normalised SCM, where we train only the first layer and fix the second-layer weights at $v_m^* = 1/M$ and $v_k = 1/K$. The asymptotic error of normalised SCM decreases with $K$ (orange curve in Fig. 4), because the second-layer weights $v_k = 1/K$ effectively reduce the learning rate, as can be easily seen from the SGD updates (2), and we know from our analysis of linear SCM in Sec. 2 that $\epsilon_g \sim \eta$. In SM Sec. F we show analytically how imbalance in the norms of the first and second layer weights can lead to a larger effective learning rate. Normalised SCM also beat the performance students where we trained both layers, starting from small initial weights in both cases. This is surprising because we checked experimentally that the weights of a normalised SCM after

training are a fixed point of the SGD dynamics when training both layers. However, we confirmed experimentally that SGD does not find this fixed point when starting with random initial weights.

**Discussion.** The qualitative difference between training both or only the first layer of neural networks is particularly striking for linear networks, where fixing one layer does not change the class of functions the model can implement, but makes a dramatic difference for their asymptotic performance. This observation highlights two important points: first, the performance of a network is not just determined by the number of additional parameters, but also by how the additional parameters are arranged in the model. Second, the non-linear dynamics of SGD means that changing which weights are trainable can alter the training dynamics in unexpected ways. We saw this for two-layer linear networks, where SGD did not find the optimal fixed point, and in the non-linear sigmoidal networks, where training the second layer allowed the student to decrease its final error with every additional hidden unit instead of increasing it like in the SCM.

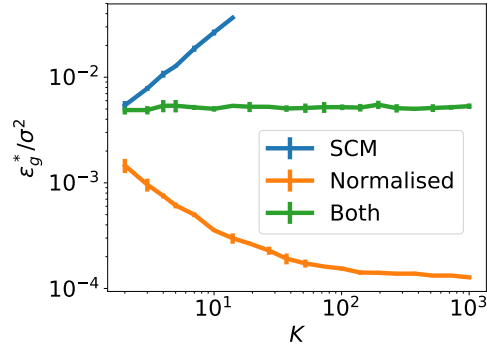

Figure 4: Asymptotic performance of linear two layer network. Error bars indicate one standard deviation over five runs. Parameters: $N = 100, M = 4, v^* = 1, \eta = 0.01, \sigma = 0.01$.

# Acknowledgements

SG and LZ acknowledge funding from the ERC under the European Union's Horizon 2020 Research and Innovation Programme Grant Agreement 714608-SMiLe. MA thanks the Swartz Program in Theoretical Neuroscience at Harvard University for support. AS acknowledges funding by the European Research Council, grant 725937 NEUROABSTRACTION. FK acknowledges support from "Chaire de recherche sur les modèles et sciences des données", Fondation CFM pour la Recherche-ENS, and from the French National Research Agency (ANR) grant PAIL.

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
