[Supplementary Material]

Supplemental Material

# Dynamics of stochastic gradient descent for two-layer neural networks in the teacher-student setup

## Contents

## A. Proof of Theorem 1.1

### A.1. Outline

We will prove Theorem 1.1 in two steps. First, we will show that the mean values of the order parameters $R_{in}$, $Q_{ik}$ and $v_k$ are given by the expressions used in the equations of motion (Lemma A.1) and that they concentrate, *i.e.* that their variance is bounded by a term of order $N^{-2}$. This ensures that the leading-order of the average increment is captured by the ODE of Theorem 1.1, and that the stochastic part of the increment of the order parameters can be ignored in the thermodynamic limit $N \to \infty$. In other words, the two bounds ensure that the stochastic Markov process converges to a deterministic process. To complete the proof, we use a form of the coupling trick as described by Wang et al. [1].

### A.2. First moments of the increment $m^\mu$

Throughout this paper, we use the convention that $\mathbb{E}$ indicates an average over all the random variables that follow, while $\mathbb{E}_\mu$ denotes the conditional expectation of all the random variables that follow *conditioned* on the state of the Markov chain at step $\mu$, $m^\mu$.

**Lemma A.1.** *Under the same setting as Theorem 1.1, for all $\mu < NT$, we have*

$$\mathbb{E} \left| \mathbb{E}_\mu \, m^{\mu+1} - m^\mu - \frac{1}{N} f(m^\mu) \right| \leq C N^{-3/2}. \tag{S1}$$

*Proof.* We first recall that $m^\mu$ contains all time-dependent order parameters $R^\mu$, $Q^\mu$, and $v^\mu$, so we will prove the Lemma in turn for each of them. In fact, in each case we can prove a slightly stronger result which encompasses the required bound.

For the *teacher-student* overlaps $R_{in}^\mu$, we multiply the update (2) with $w_n^*/N$ on both sides and find that

$$R_{in}^{\mu+1} = R_{in}^\mu - \frac{\eta_w}{N} v_i \rho_n^\mu g'(\lambda_i^\mu) \Delta^\mu. \tag{S2}$$

The local field of the teacher is $\rho_n^\mu \equiv w_n^* x^\mu / \sqrt{N}$ is a Gaussian random variable with mean zero and variance $T_{nn}$. Taking the conditional expectation, we find

$$\mathbb{E}_\mu \, R_{in}^{\mu+1} - R_{in}^\mu = \frac{1}{N} \eta_w v_i \langle \rho_n^\mu \Delta^\mu g'(\lambda_i^\mu) \rangle \tag{S3}$$

as required.

For the *student-student* overlaps $Q_{ik}^\mu$, we multiply the update (2) by $w_k^\mu/N$ and find that

$$\begin{aligned}
Q_{ik}^{\mu+1} = Q_{ik}^\mu &- \frac{1}{N} \left( \eta_w \Delta^\mu v_k^\mu g'(\lambda_k^\mu) \lambda_i^\mu + \eta_w \Delta^\mu v_i^\mu g'(\lambda_i^\mu) \lambda_k^\mu \right) \\
&+ \frac{1}{N} \left( \eta_w^2 (\Delta^\mu)^2 v_i^\mu v_k^\mu g'(\lambda_i^\mu) g'(\lambda_k^\mu) \frac{(x^\mu)^2}{N} \right).
\end{aligned} \tag{S4}$$

Using assumption (A1), we see that the term $(x^\mu)^2/N$ concentrates to yield 1 by the central limit theorem. Thus we find after taking the conditional expectation of both sides and using $\mathbb{E}_\mu \zeta^\mu = 0$ that

$$\mathbb{E}_\mu \, Q_{ik}^{\mu+1} - Q_{ik}^\mu = \frac{1}{N} f_Q(m^\mu). \tag{S5}$$

Finally, it is easy to convince oneself that taking the conditional expectation of the update for the *second-layer weights* (3) yields

$$\mathbb{E}_\mu \, v_k^{\mu+1} - v_k^\mu = \frac{1}{N} f_v(m^\mu) \tag{S6}$$

which completes the proof of Lemma A.1. $\qquad\square$

### A.3. Second moments of the increment $m^\mu$

We now proceed to bound the second-order moments of the increments of the time-dependent order parameters. We collect these bounds in the following lemma:

**Lemma A.2.** *Under the assumptions of Theorem 1.1, for all $\mu < NT$, we have that*

$$\mathbb{E} \left\| m^{\mu+1} - \mathbb{E}_\mu \, m^{\mu+1} \right\|^2 \leq C(T)N^{-2}. \tag{S7}$$

Before proceeding with the proof, we state a simple technical lemma that will be helpful in the following; we relegate its proof to Sec. A.5.

**Lemma A.3.** *Under the same assumptions as Theorem 1.1, we have for all $0 \leq \mu \leq NT$ that*

$$\mathbb{E} \, v_k^\mu \leq C(T), \tag{S8}$$

*where $C(T)$ is a constant independent of $N$.*

In the following, we will use $q$ to denote any order-parameter that is varying in time, such as the teacher-student overlaps $R_{in}^\mu$, while we keep $m^\mu$ as the collection of all order parameters, including those that are static, such as the teacher-teacher overlaps $T_{nm}$.

*Proof of Lemma A.2.* We first note all order parameters $q \in \{R_{in}, Q_{ik}, v_k\}$ obey update equations of the form

$$q^{\mu+1} = q^\mu + \frac{1}{N} f_q(m^\mu, x^\mu), \tag{S9}$$

where we have emphasised that the update function $f_q(\cdot)$ may depend on all order parameters at time $\mu$ and the $\mu$th sample shown to the student $x^\mu$. For the variance $\sigma_q^2 = \mathbb{E} \, (q - \mathbb{E} \, q)^2$ of the order parameter $q$, a little algebra yields the recursion relation

$$\left(\sigma_q^{\mu+1}\right)^2 - \left(\sigma_q^\mu\right)^2 = \frac{2}{N} \left(\mathbb{E} \, q^\mu f_q(m^\mu, x^\mu) - \mathbb{E} \, f_q(m^\mu, x^\mu) \mathbb{E} \, q^\mu\right)$$
$$+ \frac{1}{N^2} \left(\mathbb{E} \, f_q(m^\mu, x^\mu)^2 - [\mathbb{E} \, f_q(m^\mu, x^\mu)]^2\right). \tag{S10}$$

We will now use complete induction to show that for any $q$, the update of the variance at every step is bounded by $C(T)N^{-2}$ as required. In particular, this means showing that the term proportional to $N^{-1}$ actually scales as $N^{-2}$.

For the induction start, we note that by Assumption A3, we have $\sigma_q^0 = 0$. Hence the variance of any order parameter after a single step of SGD reads

$$(\sigma_q^1)^2 = \frac{2}{N} \left(\mathbb{E} \, q^0 \mathbb{E} \, f_q(m^0, x^0) - \mathbb{E} \, f_q(m^0, x^0) \mathbb{E} \, q^0\right)$$
$$+ \frac{1}{N^2} \left(\mathbb{E} \, f_q(m^0, x^0)^2 - \left[\mathbb{E} \, f_q(m^0, x^0)\right]^2\right) \tag{S11}$$
$$= \frac{1}{N^2} \left(\mathbb{E} \, f_q(m^0, x^0)^2 - \left[\mathbb{E} \, f_q(m^0, x^0)\right]^2\right). \tag{S12}$$

In going from the first to the second line, we have used assumption (A3) by which the initial macroscopic state is deterministic and therefore the average $\mathbb{E}$ is just an average over the first sample shown during training, which leads to the simplification of Eq. S12.

For the induction step, we assume that the variance after $\mu < T$ steps is $(\sigma_v^\mu)^2 \leq C(T)\mu N^{-2} \leq C(T)\alpha N^{-1}$. By using the existence and boundedness of the derivatives of the activation function, we can write $m^\mu = \mathbb{E} \, m^\mu + (m^\mu - \mathbb{E} \, m^\mu)$ and expand the terms proportional to $N^{-1}$ using a multivariate Taylor expansion in $(m^\mu - \mathbb{E} \, m^\mu)$. We find that

$$(\mathbb{E} \, q^\mu f_q(m^\mu, x^\mu) - \mathbb{E} \, f_q(m^\mu, x^\mu) \mathbb{E} \, q^\mu) \leq C(T)\mathbb{E} \, (m^\mu - \mathbb{E} \, m^\mu) \leq C(T)\sigma_q^2 \leq C(T)\sigma_q N^{-1}. \tag{S13}$$

We are justified in truncating the expansion since we assumed that $\sigma_q^2 \le C(T)N^{-1}$. If the functions $f_q(m, x)$ are bounded by a constant, this completes the induction and shows that the variance of the increment of the order parameters is bounded by $C(T)N^{-2}$, as required.

It is easy to check that all three functions $f_v$, $f_R$ and $f_Q$ fulfil this condition because of the boundedness of $g(x)$ and its derivatives (A2) and of Lemma A.3, which completes the proof of Lemma A.2. $\qquad\square$

## A.4. Putting it all together

Having proved both Lemmas A.1 and A.2, we can proceed to prove Theorem 1.1 by using the coupling trick in the form given by Wang et al. [1] for another online learning problem, namely the training of generative adversarial networks. We paraphrase the coupling trick as given by Wang et al. in the following to make the proof self-contained and refer to the supplemental material of their paper for additional details.

*Proof of Theorem 1.1.* We first define a stochastic process $b^\mu$ that is coupled with the Markov process $m^\mu$ as

$$b^{\mu+1} = b^\mu + \frac{1}{N}g(m^\mu) + m^{\mu+1} - \mathbb{E}_\mu \, m^{\mu+1}. \tag{S14}$$

This process lives in the same space as $m^\mu$. Wang et al. [1] showed that for such a process, when Lemma A.1 holds, we have that

$$\mathbb{E}\,||b^\mu - m^\mu|| \le C(T)N^{-1/2} \tag{S15}$$

for all $\mu \le NT$. We then define a deterministic process

$$d^{\mu+1} = d^\mu + \frac{1}{N}g(d^\mu), \tag{S16}$$

which is a standard first-order finite difference approximation of the equations of motion (9), and also lives in the space as $m^\mu$. Invoking a standard Euler argument for first-order finite differences gives

$$\mathbb{E}\,||d^\mu - m(\mu/N)|| \le C(T)N^{-1}. \tag{S17}$$

Wang et al. [1] further showed that for such a process, using Lemma A.2, we have

$$\mathbb{E}\,||b^\mu - d^\mu|| \le C(T)N^{-1}. \tag{S18}$$

Finally, combining Eqs. (S15), (S18) and (S17), we have

$$\mathbb{E}\,||m^\mu - m(\mu/N)|| \le C(T)N^{-1/2} \tag{S19}$$

which completes the proof. $\qquad\square$

## A.5. Additional proof details

*Proof of Lemma A.3.* The increment of $v_k$ reads explicitly

$$v_k^{\mu+1} - v_k^\mu = \frac{\eta_v}{N}\left[\sum_m v_m^* g(\rho_m^\mu) - \sum_k v_k^\mu g\left(\lambda_k^\mu\right) - \sigma\zeta^\mu\right]. \tag{S20}$$

To bound the value of $v_k^\mu$ after $\mu$ steps, we consider the three terms in the sum $v_k^\mu = \sum_{s=1}^\mu v_k^\mu$ each in turn. We first note that the sum of the output noise variables $\zeta^\mu$ is a simple sum over uncorrelated, (sub-) Gaussian random variables rescaled by $1/N$ and thus by Hoeffding's inequality almost surely smaller than a constant [2].

For the first two terms, we can use an argument similar to the one used to prove the bound on the variance of the increment of the order parameters. We first note that $g(\cdot)$ is a bounded

function by Assumption (A2) and that the initial conditions of the second-layer weights are bounded by a constant by Assumption (A3). Hence, after a first step, the weight has increased by a term bounded by $C(T)N^{-1}$. Actually, at every step where the weight is bounded by a constant, its increase will be bounded by $C(T)N^{-1}$. Hence the magnitude of $v_k^\mu \leq C(T)$ for $0 \leq \mu \leq NT$, as required. $\qquad\square$

## B. Derivation of the ODE description of the generalisation dynamics of online learning

Here we demonstrate how to evaluate the averages found in the equations of motion for the order parameters (9), following the classic work by Biehl and Schwarze [3] and Saad and Solla [4, 5]. We repeat the two main technical assumption of our work, namely having a large network ($N \to \infty$) and a data set that is large enough to allow that we visit every sample only once before training converges. Both will play a key role in the following computations.

### B.1. Expressing the generalisation error in terms of order parameters

We first demonstrate how the assumptions stated above allow to rewrite the generalisation error in terms of a number of *order parameters*. We have

$$\epsilon_g(\theta, \theta^*) \equiv \frac{1}{2} \left\langle [\phi(x, \theta) - \phi(x, \theta^*)]^2 \right\rangle \tag{S21}$$

$$= \frac{1}{2} \left\langle \left[ \sum_{k=1}^{K} v_k g(\lambda_k) - \sum_{m=1}^{M} v_m^* g(\rho_m) \right]^2 \right\rangle, \tag{S22}$$

where we have used the local fields $\lambda_k$ and $\rho_m$. Here and throughout this paper, we will use the indices $i, j, k, \ldots$ to refer to hidden units of the student, and indices $n, m, \ldots$ to denote hidden units of the teacher. Since the input $x^\mu$ only appears in $\epsilon_g$ only via products with the weights of the teacher and the student, we can replace the high-dimensional average $\langle \cdot \rangle$ over the input distribution $p(x)$ by an average over the $K + M$ local fields $\lambda_k^\mu$ and $\rho_m^\mu$. The assumption that the training set is large enough to allow that we visit every sample in the training set only once guarantees that the inputs and the weights of the networks are uncorrelated. Taking the limit $N \to \infty$ ensures that the local fields are jointly normally distributed with mean zero ($\langle x_n \rangle = 0$). Their covariance is also easily found: writing $w_{ka}$ for the $a$th component of the $k$th weight vector, we have

$$\langle \lambda_k \lambda_l \rangle = \frac{\sum_{a,b}^{N} w_{ka} w_{lb} \langle x_a x_b \rangle}{N} = \frac{w_k w_l}{N} \equiv Q_{kl}, \tag{S23}$$

since $\langle x_a x_b \rangle = \delta_{ab}$. Likewise, we define

$$\langle \rho_n \rho_m \rangle = \frac{w_n^* w_m^*}{N} \equiv T_{nm}, \quad \langle \lambda_k \rho_m \rangle = \frac{w_k w_m^*}{N} \equiv R_{km}. \tag{S24}$$

The variables $R_{in}$, $Q_{ik}$, and $T_{nm}$ are called *order parameters* in statistical physics and measure the overlap between student and teacher weight vectors $w_i$ and $w_n^*$ and their self-overlaps, respectively. Crucially, from Eq. (S22) we see that they are sufficient to determine the generalisation error $\epsilon_g$. We can thus write the generalisation error as

$$\epsilon_g = \frac{1}{2} \sum_{i,k} v_i v_k I_2(i, k) + \frac{1}{2} \sum_{n,m} v_n^* v_m^* I_2(n, m) - \sum_{i,n} v_i v_n^* I_2(i, n), \tag{S25}$$

where we have defined

$$I_2(i, k) \equiv \langle g(\lambda_i) g(\lambda_k) \rangle. \tag{S26}$$

Assuming sigmoidal activation functions $g(x) = \mathrm{erf}(x/\sqrt{2})$ allows us to evaluate the average $I_2(i,k)$ analytically:

$$I_2(i,k) = \frac{1}{\pi} \arcsin \frac{Q_{ik}}{\sqrt{1+Q_{ii}}\sqrt{1+Q_{kk}}}. \tag{S27}$$

The average in Eq. (S26) is taken over a normal distribution for the local fields $\lambda_i$ and $\lambda_k$ with mean $(0,0)$ and covariance matrix

$$C_2 = \begin{pmatrix} Q_{ii} & Q_{ik} \\ Q_{ik} & Q_{kk} \end{pmatrix}. \tag{S28}$$

Since we are using the indices $i, j, \ldots$ for student units and $n, m, \ldots$ for teacher hidden units, we have

$$I_2(i,n) = \langle g(\lambda_i) g(\rho_n) \rangle, \tag{S29}$$

where the covariance matrix of the joint of distribution $\lambda_i$ and $\rho_m$ is given by

$$C_2 = \begin{pmatrix} Q_{ii} & R_{in} \\ T_{in} & T_{nn} \end{pmatrix}. \tag{S30}$$

and likewise for $I_2(n,m)$. We will use this convention to denote integrals throughout this section. For the generalisation error, this means that it can be expressed in terms of the order parameters alone as

$$\epsilon_g = \frac{1}{\pi} \sum_{i,k} v_i v_k \arcsin \frac{Q_{ik}}{\sqrt{1+Q_{ii}}\sqrt{1+Q_{kk}}} + \frac{1}{\pi} \sum_{n,m} v_n^* v_m^* \arcsin \frac{T_{nm}}{\sqrt{1+T_{nn}}\sqrt{1+T_{mm}}}$$
$$- \frac{2}{\pi} \sum_{i,n} v_i v_n^* \arcsin \frac{R_{in}}{\sqrt{1+Q_{ii}}\sqrt{1+T_{nn}}}. \tag{S31}$$

## B.2. ODEs for the evolution of the order parameters

Expressing the generalisation error in terms of the order parameters as we have in Eq. (S31) is of course only useful if we can track the evolution of the order parameters over time. We can derive ODEs that allow us to do precisely that for the order parameters $Q$ by squaring the weight update of $w$ (2) and for $R$ taking the inner product of (2) with $w_n^*$, respectively, which yields the equations of motion (9).

To make progress however, *i.e.* to obtain a closed set of differential equations for $Q$ and $R$, we need to evaluate the averages $\langle \cdot \rangle$ over the local fields. In particular, we have to compute three types of averages:

$$I_3 = \langle g'(a) b g(c) \rangle, \tag{S32}$$

where $a$ is one the local fields of the student, while $b$ and $c$ can be local fields of either the student or the teacher;

$$I_4 = \langle g'(a) g'(b) g(c) g(d) \rangle, \tag{S33}$$

where $a$ and $b$ are local fields of the student, while $c$ and $d$ can be local fields of both; and finally

$$J_2 = \langle g'(a) g'(b) \rangle, \tag{S34}$$

where $a$ and $b$ are local fields of the teacher. In each of these integrals, the average is taken with respect to a multivariate normal distribution for the local fields with zero mean and a covariance matrix whose entries are chosen in the same way as discussed for $I_2$.

We can re-write Eqns. (9) with these definitions in a more explicit form as [4–6]

$$\frac{\mathrm{d}R_{in}}{\mathrm{d}t} = \eta v_i \left[ \sum_m^M v_m^* I_3(i,n,m) - \sum_j^K v_j I_3(i,n,j) \right], \tag{S35}$$

$$\frac{\mathrm{d}Q_{ik}}{\mathrm{d}t} = \eta_w v_i \left[ \sum_m^M v_m^* I_3(i,k,m) - \sum_j^K v_j I_3(i,k,j) \right]$$

$$+ \eta_w v_k \left[ \sum_m^M v_m^* I_3(k,i,m) - \sum_j^K v_j I_3(i,j,k) \right]$$

$$+ \eta_w^2 v_i v_k \left[ \sum_n^M \sum_m^M v_n^* v_m^* I_4(i,k,n,m) - 2 \sum_j^K \sum_n^M v_j v_n^* I_4(i,k,j,n) \right.$$

$$\left. + \sum_j^K \sum_l^K v_j v_l I_4(i,k,j,l) + \sigma^2 J_2(i,k) \right] \tag{S36}$$

$$\frac{\mathrm{d}v_i}{\mathrm{d}t} = \eta_v \left[ \sum_n^M v_n^* I_2(i,n) - \sum_j^K v_j I_2(i,j) \right] . \tag{S37}$$

The explicit form of the integrals $I_2(\cdot)$, $I_3(\cdot)$, $I_4(\cdot)$ and $J_2(\cdot)$ is given in Sec. H for the case $g(x) = \mathrm{erf}\left(x/\sqrt{2}\right)$. Solving these equations numerically for $Q$ and $R$ and substituting their values in to the expression for the generalisation error (S25) gives the full generalisation dynamics of the student. We show the resulting learning curves together with the result of a single simulation in Fig. 2 of the main text. We have bundled our simulation software and our ODE integrator as a user-friendly library with example programs at `https://github.com/sgoldt/nn2pp`. In Sec. C, we discuss how to extract information from them in an analytical way.

## C. Calculation of $\epsilon_g$ in the limit of small noise for Soft Committee Machines

Our aim is to understand the asymptotic value of the generalisation error

$$\epsilon_g^* \equiv \lim_{\alpha \to \infty} \epsilon_g(\alpha). \tag{S38}$$

We focus on students that have more hidden units than the teacher, $K \geq M$. These students are thus over-parameterised *with respect to the generative model of the data* and we define

$$L \equiv K - M \tag{S39}$$

as the number of additional hidden units in the student network. In this section, we focus on the sigmoidal activation function

$$g(x) = \mathrm{erf}\left(x/\sqrt{2}\right), \tag{S40}$$

unless stated otherwise.

Eqns. (S35ff) are a useful tool to analyse the generalisation dynamics and they allowed Saad and Solla to gain plenty of analytical insight into the special case $K = M$ [4, 5]. However, they are also a bit unwieldy. In particular, the number of ODEs that we need to solve grows with $K$ and $M$ as $K^2 + KM$. To gain some analytical insight, we make use of the symmetries in the problem, *e.g.* the permutation symmetry of the hidden units of the student, and re-parametrised the matrices $Q_{ik}$ and $R_{in}$ in terms of eight order parameters that obey a set of self-consistent

ODEs for any $K > M$. We choose the following parameterisation with eight order parameters:

$$
Q_{ij} = \begin{cases}
Q & i = j \leq M, \\
C & i \neq j;\ i, j \leq M, \\
D & i > M, j \leq M \quad \text{or} \quad i \leq M, j > M, \\
E & i = j > M, \\
F & i \neq j;\ i, j > M,
\end{cases}
\tag{S41}
$$

$$
R_{in} = \begin{cases}
R & i = n, \\
S & i \neq n;\ i \leq M, \\
U & i > M,
\end{cases}
\tag{S42}
$$

which in matrix form for the case $M = 3$ and $K = 5$ read:

$$
R = \begin{pmatrix}
R & S & S \\
S & R & S \\
S & S & R \\
U & U & U \\
U & U & U
\end{pmatrix}
\quad \text{and} \quad
Q = \begin{pmatrix}
Q & C & C & D & D \\
C & Q & C & D & D \\
C & C & Q & D & D \\
D & D & D & E & F \\
D & D & D & F & E
\end{pmatrix}.
\tag{S43}
$$

We choose this number of order parameters and this particular setup for the overlap matrices $Q$ and $R$ for two reasons: it is the smallest number of variables for which we were able to self-consistently close the equations of motion (S35), and they agree with numerical evidence obtained from integrating the full equations of motion (S35).

By substituting this ansatz into the equations of motion (S35), we find a set of eight ODEs for the order parameters. These equations are rather unwieldy and some of them do not even fit on one page, which is why we do not print them here in full; instead, we provide a *Mathematica* notebook where they can be found and interacted with together with the source at `http://www.github.com/sgoldt/nn2pp`. These equations allow for a detailed analysis of the effect of over-parameterisation on the asymptotic performance of the student, as we will discuss now.

### C.1. Heavily over-parameterised students can learn perfectly from a noiseless teacher using online learning

For a teacher with $T_{nm} = \delta_{nm}$ and in the absence of noise in the teacher's outputs ($\sigma = 0$), there exists a fixed point of the ODEs with $R = Q = 1$, $C = D = E = F = 0$, and perfect generalisation $\epsilon_g = 0$. Online learning will find this fixed point [4, 5]. More precisely, after a plateau whose length depends on the size of the network for the sigmoidal network, the generalisation error eventually begins an exponential decay to the optimal solution with zero generalisation error. The learning rates are chosen such that learning converges, but aren't optimised otherwise.

### C.2. Perturbative solution of the ODEs

We have calculated the asymptotic value of the generalisation error $\epsilon_g^*$ for a teacher with $T_{nm} = \delta_{nm}$ to first order in the variance of the noise $\sigma^2$. To do so, we performed a perturbative expansion around the fixed point

$$
R_0 = Q_0 = 1,
\tag{S44}
$$

$$
S_0 = U_0 = C_0 = D_0 = E_0 = F_0 = 0,
\tag{S45}
$$

with the ansatz

$$
X = X_0 + \sigma^2 X_1
\tag{S46}
$$

**Figure S1: The final generalisation error of over-parameterised sigmoidal networks scales linearly with the learning rate, the variance of the teacher's output noise, and $L$.** We plot $\epsilon_g^*/\sigma^2$ in the limit of small noise, Eq. (S47), for $M = 2$ (red) and $M = 16$ (blue). It is clear that generalisation error increases with the number of superfluous units $L$ at fixed learning rate (*left*) and the learning rate $\eta$ (*middle*). *Right:* For $K = M$, the learning rate $\eta_{\text{div}}$ at which our perturbative result diverges is precisely the maximum learning rate $\eta_{\text{max}}$ at which the exponential convergence to the optimal solution is guaranteed for $\sigma = 0$, Eq. (S48)

for all the order parameters. Writing the ODEs to first order in $\sigma^2$ and solving for their steady state where $X'(\alpha) = 0$ yielded a fixed point with an asymptotic generalisation error

$$\epsilon_g^* = \frac{\sigma^2 \eta}{2\pi} f(M, L, \eta) + \mathcal{O}(\sigma^3). \tag{S47}$$

$f(M, L, \eta)$ is an unwieldy rational function of its variables. Due to its length, we do not print it here in full; instead, we give the full function in a *Mathematica* notebook together with our source code at `https://github.com/anon/...`. Here, we plot the results in various forms in Fig. S1. We note in particular the following points:

$\epsilon_g^*$ **increases with $L$, $\eta$** The two plots on the left show that the generalisation error increases monotonically with both $L$ and $\eta$ while keeping the other fixed, respectively, for teachers with $M = 2$ (red) and $M = 16$ (blue)

**The role of the learning rate $\eta$** Mitigating this effect by decreasing the learning rate $\eta$ for larger students raises two problems: a lower learning rate yields longer training times, and longer training times imply that more data is required until convergence. This is in agreement with statistical learning theory, where models with more parameters generalise just as well as smaller ones given enough data, despite having a higher complexity class as measured by VC dimension or Rademacher complexity [7], for example. Furthermore, we show in Sec. C.2 that even with $\eta \sim 1/K$, the generalisation error increases with $L$ before plateauing at a constant value. Moreover, we show in Fig. S2 that the asymptotic generalisation error (S47) of a student trained using SGD with learning rate $\eta = 1/K$ still increases with $L$ before plateauing at a constant value that is independent of $M$.

**Divergence at large $\eta$** Our perturbative result diverges for large $L$, or equivalently, for a large learning rate that depends on the number of hidden units $L \sim K$. For the special case $K = M$, the learning rate $\eta_{\text{div}}$ at which our perturbative result diverges is precisely the maximum learning rate $\eta_{\text{max}}$ for which the exponential convergence to the optimal solution is still guaranteed for $\sigma = 0$ [5]

$$\eta_{\text{max}} = \frac{\sqrt{3}\pi}{M + 3/\sqrt{5} - 1} \tag{S48}$$

as we show in the right-most plot of Fig. S1.

**Expansion for small $\eta$** In the limit of small learning rates, which is the most relevant in practice and which from the plots in Fig. S1 dominates the behaviour of $\epsilon_g^*$ outside of the divergence,

**Figure S2: Asymptotic generalisation error for sigmoidal soft committee machines with learning rate** $\eta = 1/K$**.** We plot the asymptotic generalisation error $\epsilon_g^*$ (S47) over $\sigma^2$ of a student with a varying number of hidden units trained on data generated by teachers with $M = 2, 4, 16$ using SGD with learning rate $1/K$. The generalisation error still increases with $K$, before plateauing at a constant value that is independent of $M$. Weight decay parameter $\kappa = 0$.

the generalisation error is linear in the learning rate. Expanding $\epsilon_g^*$ to first order in the learning rate reveals a particularly revealing form,

$$\epsilon_g^* = \frac{\sigma^2 \eta}{2\pi} \left( L + \frac{M}{\sqrt{3}} \right) + \mathcal{O}(\eta^2) \tag{S49}$$

with second-order corrections that are quadratic in $L$. This is actually the sum of the asymptotic generalisation errors of $M$ continuous perceptrons that are learning from a teacher with $T = 1$ and $L$ continuous perceptrons with $T = 0$ as we calculate in Sec. D. This neat result is a consequence of the specialisation that is typical of SCMs with sigmoidal activation functions as we discussed in the main text.

## D. Asymptotic generalisation error of a noisy continuous perceptron

What is the asymptotic generalisation for a continuous perceptron, *i.e.* a network with $K = 1$, in a teacher-student scenario when the teacher has some additive Gaussian output noise? In this section, we repeat a calculation by Biehl and Schwarze [3] where the teacher's outputs are given by

$$y = g \left( \frac{w^* x}{\sqrt{N}} \right) + \zeta, \tag{S50}$$

where $\zeta$ is again a Gaussian r.v. with mean 0 and variance $\sigma^2$. We keep denoting the weights of the student by $w$ and the weights of the teacher by $w^*$. To analyse the generalisation dynamics, we introduce the order parameters

$$R \equiv \frac{w w^*}{N}, \qquad Q \equiv \frac{w w}{N} \quad \text{and} \quad T \equiv \frac{w^* w^*}{N}. \tag{S51}$$

**Figure S3: The final generalisation error of over-parametrised ReLU networks scales as $\epsilon_g^* \sim \eta\sigma^2 L$.** Simulations confirm that the asymptotic generalisation error $\epsilon_g^*$ of a ReLU student learning from a ReLU teacher scales with the learning rate $\eta$, the variance of the teacher's output noise $\sigma^2$ and the number of additional hidden units as $\epsilon_g \sim \eta\sigma^2 L$, which is the same scaling as the one found analytically for sigmoidal networks in Eq. (S49). Straight lines are linear fits to the data, with slope 1 in (a) and (b). Parameters: $M = 2, K = 6$ (a, b) and $M = 4, 16; K = M + L$ (c); in all figures, $N = 784, \kappa = 0$.

and we explicitly do not fix $T$ for the moment. For $g(x) = \mathrm{erf}\left(x/\sqrt{2}\right)$, they obey the following equations of motion:

$$\frac{dR}{dt} = \frac{2\eta}{\pi\left(Q(t)+1\right)}\left(\frac{TQ(t)-R(t)^2+T}{\sqrt{(T+1)Q(t)-R(t)^2+T+1}} - \frac{R(t)}{\sqrt{2Q(t)+1}}\right) \tag{S52}$$

$$\frac{dQ}{dt} = \frac{4\eta}{\pi(Q(t)+1)}\left(\frac{R(t)}{\sqrt{2(Q(t)+1)-R(t)^2}} - \frac{Q(t)}{\sqrt{2Q(t)+1}}\right)$$

$$+ \frac{4\eta^2}{\pi^2\sqrt{2Q(t)+1}}\left[-2\arcsin\left(\frac{R(t)}{\sqrt{(6Q(t)+2)(2Q(t)-R(t)^2+1)}}\right)\right.$$

$$\left. + \arcsin\left(\frac{2\left(Q(t)-R(t)^2\right)+1}{2\left(2Q(t)-R(t)^2+1\right)}\right) + \arcsin\left(\frac{Q(t)}{3Q(t)+1}\right)\right]$$

$$+ \frac{2\eta^2\sigma^2}{\pi\sqrt{2Q(t)+1}}. \tag{S53}$$

The equations of motion have a fixed point at $Q = R = T$ which has perfect generalisation for $\sigma = 0$. We hence make a perturbative ansatz in $\sigma^2$

$$Q(t) = T + \sigma^2 q(t) \tag{S54}$$
$$R(t) = T + \sigma^2 r(t) \tag{S55}$$

and find for the asymptotic generalisation error

$$\epsilon_g^* = \frac{\eta\sigma^2(4T+1)}{2\sqrt{2T+1}\left(-\eta\sqrt{8T^2+6T+1}+4\pi T+\pi\right)} + \mathcal{O}\left(\sigma^3\right). \tag{S56}$$

To first order in the learning rate, this reads

$$\epsilon_g^* = \frac{\eta\sigma^2}{2\pi\sqrt{2T+1}}, \tag{S57}$$

which should be compared to the corresponding result for the full SCMs, Eq. (S49).

**Figure S4:** Asymptotic performance of linear (left) and ReLU (right) two layer networks. Error bars indicate one standard deviation over five runs, and the y-axis is the same in both plots. Parameters: $N = 500, M = 2, v^* = 4, \eta = 0.01, \sigma = 0.01$. N.B. The right plot is the same as Fig. 4 of the main text.

## E. Calculation of the asymptotic generalisation error in two-layer sigmoidal networks

In this section, we describe the ansatz we chose for the ODE to compute the asymptotic generalisation error when training both layers with sigmoidal activation function. As we describe in the main text, the ansatz used for the Soft Committee Machine is not appropriate, since (i) all the hidden units of the student are used, and (ii) several nodes overlap with the same teacher node. Inspection of the overlaps after integration of the ODE thus suggested the following ansatz when the number of nodes in the student is a multiple of the number of teacher nodes, $K = ZM$:

$$Q_{ij} = \begin{cases} Q & i \mod M = j \mod M, \\ C & \text{otherwise} \end{cases} \tag{S58}$$

$$R_{in} = \begin{cases} R & i \mod M = n \mod M, \\ S & \text{otherwise} \end{cases} \tag{S59}$$

which in matrix form for the case $M = 2$ and $K = 4$ read:

$$R_{in} = \begin{pmatrix} R & S \\ S & R \\ R & S \\ S & R \end{pmatrix} \quad \text{and} \quad Q_{ik} = \begin{pmatrix} Q & C & Q & C \\ C & Q & C & Q \\ Q & C & Q & C \\ C & Q & C & Q \end{pmatrix} \tag{S60}$$

Once this ansatz is found, the rest of the calculation follows along the same lines as that of Sec. C: we derive a reduced set of coupled ODE for $Q, C, R$ and $S$, expand around the noiseless fixed point where $R = 1, S = 0, Q = 1, C = 0$ and substitute the resulting fixed point into the expression for the generalisation error, yielding the formula plotted in Fig. 3c.

In Fig. S4 we show the asymptotic performance linear and ReLU two-layer networks that we discuss at the end of Sec. 3 of the main text.

## F. Unbalanced weights rescale effective learning rate in two layer linear networks

If we consider a linear, two layer neural network of the form:

$$\phi(x, \theta) = \sum_{m,j} v_m w_{mj} x_j, \tag{S61}$$

where $v \in \mathbb{R}^{1 \times M}$, $w \in \mathbb{R}^{M \times N}$ and $x \in \mathbb{R}^{N \times 1}$. The online SGD updates to the first and second layer weights will have the form:

$$\Delta w_{mj}^{\mu} = \eta(y^{\mu} - \phi(x^{\mu}, \theta^{\mu}))v_m^{\mu} x_j^{\mu}, \tag{S62}$$

and

$$\Delta v_m^{\mu} = \eta(y^{\mu} - \phi(x^{\mu}, \theta^{\mu})) \sum_j w_{mj} x_j^{\mu}. \tag{S63}$$

If we define the product of student weights as a vector $u$:

$$u_j = \sum_{m=1}^{M} v_m w_{mj}, \tag{S64}$$

it follows that

$$\Delta u_j^{\mu} = \sum_{m=1}^{M} \left( v_m^{\mu} \Delta w_{mj}^{\mu} + \Delta v_m^{\mu} w_{mj}^{\mu} \right). \tag{S65}$$

Substituting the form for the update in first and second layer weights into the expression above we find:

$$\Delta u^{\mu} = \eta(y^{\mu} - u^{\mu} \cdot x^{\mu})(x^{\mu})^T \left( \mathbb{I}_N \|v^{\mu}\|^2 + (w^{\mu})^T(w^{\mu}) \right). \tag{S66}$$

This suggests that the level of imbalance between the norm of weights at different layers may impact the noisy fluctuations in updates even at late training times. If we compare the update step of the network with another network which produces the same output but has a different scaling of the weights we can see that the effective learning rate will be different. For instance $\tilde{v} = av$ and $\tilde{w} = \frac{1}{a}w$ leads to an equivalent network, but updates which scale as:

$$\Delta u^{\mu} = \eta(y^{\mu} - u^{\mu} \cdot x^{\mu})(x^{\mu})^T \left( \mathbb{I}_N a^2 \|v^{\mu}\|^2 + \frac{1}{a^2}(w^{\mu})^T(w^{\mu}) \right). \tag{S67}$$

We can think of this scaling of the weights as impacting the effective learning, and we have provided an expression for how the learning rate impacts generalisation error in this paper. Our finding thus suggests that weights with more balanced norms across layers will tend to lead to lower generalisation error during online learning.

## G. Additional experiments on Soft Committee Machines

### G.1. Regularisation by weight decay does not help

A natural strategy to avoid the pitfalls of overfitting is to regularise the weights, for example by using explicit weight decay by choosing $\kappa > 0$. We have not found a setup where adding weight decay *improved* the asymptotic generalisation error of a student compared to a student that was trained without weight decay in our setup. As a consequence, weight decay completely fails to mitigate the increase of $\epsilon_g^*$ with $L$. We show the results of an illustrative experiment in Fig. S5.

### G.2. SGD with mini-batches

One key characteristic of online learning is that we evaluate the gradient of the loss function using a single sample from the training step per step. In practice, it is more common to actually use a number of samples $b > 1$ to estimate the gradient at every step. To be more precise, the weight update equation for SGD with mini-batches would read:

$$w_k^{\mu+1} = w_k^{\mu} - \frac{\kappa}{N}w_k^{\mu} - \frac{\eta}{b\sqrt{N}} \sum_{\ell=1}^{b} x^{\mu,\ell} g'(\lambda_k^{\mu,\ell}) \left[ \phi(x^{\mu,\ell}, \theta) - y^{\mu,\ell} \right]. \tag{S68}$$

**Figure S5: Weight decay.** We plot the final generalisation error $\epsilon_g^*$ of a student with a varying number of hidden units trained on data generated by a teacher with $M = 4$ using SGD with weight decay. The generalisation error clearly increases with the weight decay constant $\kappa$. Parameters: $N = 784, \eta = 0.1, \sigma = 0.01$.

**Figure S6: SGD with mini-batches shows the same qualitative behaviour as online learning** We show the asymptotic generalisation error $\epsilon_g^*$ for students with sigmoidal (left) and ReLU activation function (right) for various $K$ learning from a teacher with $M = 4$. Between the curves, we change the size of the mini-batch used at each step of SGD from 1 (online learning) to 20 000. Parameters: $N = 500, \eta = 0.2, \sigma = 0.1, \kappa = 0$.

**Figure S7: Higher-order correlations in the input data do not play a role for the asymptotic generalisation.** We plot the final generalisation error $\epsilon_g^*$ after online learning of a student of various sizes from a teacher with $M = 4$ using Gaussian inputs (blue) and MNIST images (red) for training and testing. $N = 784, \eta = 0.1, \sigma = 0.1, \kappa = 0$.

where $x^{\mu,\ell}$ is the $\ell$th input from the mini-batch used in the $m$th step of SGD, $\lambda_k^{\mu,\ell}$ is the local field of the $k$th student unit for the $\ell$th sample in the mini-batch, etc. Note that when we use every sample only once during training, using mini-batches of size $b$ increases the amount of data required by a factor $b$ when keeping the number of steps constant.

We show the asymptotic generalisation error of student networks of varying size trained using SGD with mini-batches and a teacher with $M = 4$ in Fig. S6. Two trends are visible: first, using increasing the size of the mini-batches decreases the asymptotic generalisation error $\epsilon_g^*$ up to a certain mini-batch size, after which the gains in generalisation error become minimal; and second, the shape of the $\epsilon_g^* - L$ curve is the same for all mini-batch sizes, with the minimal generalisation error attained by a network with $K = M$.

## G.3. Using MNIST images for training and testing

In the derivation of the ODE description of online learning for the main text, we noted that only the first two moments of the input distribution matter for the learning dynamics and for the final generalisation error. The reason for this is that the inputs only appear in the equations of motion for the order parameters as a product with the weights of either the teacher or the student. Now since they are – by assumption – uncorrelated with those weights, this product is the sum of large number of random variables and hence distributed by the central limit theorem.

We have checked how our results change when this assumption breaks down in one example where we train a network on a finite data set with non-trivial higher order moments, namely the images of the MNIST data set. We studied the very same setup that we discuss throughout this work, namely the supervised learning of a regression task in the teacher-student scenario. We *only* replace the the inputs, which would have been i.i.d. draws from the standard normal distribution, with the images of the MNIST data set. In particular, this means that we do not care about the labels of the images. Figure S7 shows a plot of the resulting final generalisation against $L$ for both the MNIST data set and a data set of the same size, comprised of i.i.d. draws from the standard normal distribution, which are in good agreement.

## G.4. The scaling of $\epsilon_g^*$ with $L$ for finite training sets

In practice, a single sample of the training data set will be visited several times during training. After a first pass through the training set, the online assumption that an incoming sample $(x, y)$

**Figure S8: The scaling of $\epsilon_g^*$ with $L$ shows a similar dependence on the size of the training set for early-stopping (top) and final (bottom) generalisation error.** We plot the asymptotic and the early-stopping generalisation error after SGD with a finite training set containing $PN$ samples (linear, sigmoidal and ReLU networks from left to right). The result for online learning for linear and sigmoidal networks, Eqns. (10) and (12) of the main text, are plotted in violet. Error bars indicate one standard deviation over 10 simulations, each with a different training set; many of them are too small to be clearly visible. Parameters: $N = 784, M = 4, \eta = 0.1, \sigma = 0.01$.

is uncorrelated to the weights of the network thus breaks down. A complete analytical treatment in this setting remains an open problem, so to study this practically relevant setup, we turn to simulations. We keep the setup described in Secs. 1, but simply reduce the number of samples in the training data set $P$. Our focus is again on the final generalisation error after convergence $\epsilon_g^*$ for linear, sigmoidal and ReLU networks, which we plot from left to right as a function of $L$ in Fig. S8.

Linear networks show a similar behaviour to the setup with a very large training set discussed in Sec. 2: the bigger the network, the worse the performance for both $P = 4$ and $P = 50$. Again, the optimal network has $K = 1$ hidden units, irrespective of the size of the teacher. However, for non-linear networks, the picture is more varied: For large training sets, where the number of samples easily outnumber the free parameters in the network ($P = 50$, red curve; this corresponds roughly to learning a data set of the size of MNIST), the behaviour is qualitatively described by our theory from Sec. 2: the best generalisation is obtained by a network that matches the teacher size, $K = M$. However, as we reduce the size of the training set, this is no longer true. For $P = 4$, for example, the best generalisation is obtained with networks that have $K > M$. Thus the size of the training set with respect to the network has an important influence on the scaling of $\epsilon_g^*$ with $L$. Note that the early-stopping generalisation error, which we define as the minimal generalisation error over the duration of training, shows qualitatively the same behaviour as $\epsilon_g^*$.

### G.5. Early-stopping generalisation error for finite training sets

A common way to prevent over-fitting of a neural network when training with a finite training set in practice is early stopping, where the training is stopped before the training error has converged to its final value yet. The idea behind early-stopping is thus to stop training before over-fitting sets in. For the purpose of our analysis of the generalisation of two-layer networks trained on a fixed finite data set in Sec. 4 of the main text, we define the early-stopping generalisation error $\hat{\epsilon}_g$ as the minimum of $\epsilon_g$ during the whole training process. In Fig. S8, we reproduce Fig. 6 from the main text at the bottom and plot $\hat{\epsilon}_g$ obtained from the very same experiments at the top. While the ReLU networks showed very little to no over-training, the sigmoidal networks showed more significant over-training. However, the qualitative dependence of the generalisation errors on $L$ was observed to be the same in this experiment. In particular, the early-stopping generalisation error also shows two different regimes, one where increasing the network hurts generalisation ($P \gg K$), and one where it improves generalisation or at least doesn't seem to affect it much (small $P \sim K$).

## H. Explicit form of the integrals appearing in the equations of motion of sigmoidal networks

To be as self-contained as possible, here we collect the explicit forms of the integrals $I_2$, $I_3$, $I_4$ and $J_2$ that appear in the equations of motion for the order parameters and the generalisation error for networks with $g(x) = \text{erf}\left(x/\sqrt{2}\right)$, see Eq. (S35). They were first given by [3, 4]. Each average $\langle \cdot \rangle$ is taken w.r.t. a multivariate normal distribution with mean 0 and covariance matrix $C \in \mathbb{R}^n$, whose components we denote with small letters. The integration variables $u, v$ are

always components of $\lambda$, while $w$ and $z$ can be components of either $\lambda$ or $\rho$.

$$J_2 \equiv \langle g'(u)g'(v) \rangle = \frac{2}{\pi}\left(1 + c_{11} + c_{22} + c_{11}c_{22} - c_{12}^2\right)^{-1/2} \tag{S69}$$

$$I_2 \equiv \frac{1}{2}\langle g(w)g(z) \rangle = \frac{1}{\pi}\arcsin\frac{c_{12}}{\sqrt{1+c_{11}}\sqrt{1+c_{12}}}. \tag{S70}$$

$$I_3 \equiv \langle g'(u)wg(z) \rangle = \frac{2}{\pi}\frac{1}{\sqrt{\Lambda_3}}\frac{c_{23}(1+c_{11}) - c_{12}c_{13}}{1+c_{11}} \tag{S71}$$

$$I_4 \equiv \langle g'(u)g'(v)g(w)g(z) \rangle = \frac{4}{\pi^2}\frac{1}{\sqrt{\Lambda_4}}\arcsin\left(\frac{\Lambda_0}{\sqrt{\Lambda_1\Lambda_2}}\right) \tag{S72}$$

where

$$\Lambda_4 = (1+c_{11})(1+c_{22}) - c_{12}^2 \tag{S73}$$

and

$$\Lambda_0 = \Lambda_4 c_{34} - c_{23}c_{24}(1+c_{11}) - c_{13}c_{14}(1+c_{22}) + c_{12}c_{13}c_{24} + c_{12}c_{14}c_{23} \tag{S74}$$

$$\Lambda_1 = \Lambda_4(1+c_{33}) - c_{23}^2(1+c_{11}) - c_{13}^2(1+c_{22}) + 2c_{12}c_{13}c_{23} \tag{S75}$$

$$\Lambda_2 = \Lambda_4(1+c_{44}) - c_{24}^2(1+c_{11}) - c_{14}^2(1+c_{22}) + 2c_{12}c_{14}c_{24} \tag{S76}$$