[Reviews · NeurIPS 2019]

Reviewer 1



Please see comments in item 1 above. Some other comments: The statement of Thm 1.1. can be greatly improved. I suggest separating the assumptions in a previous statement, in an itemized manner. Then start the statement of the theorem with: Under Assumptions?, let $T > 0$ and $\alpha > 0$. Then, the macroscopic state ... Also, just say "where $m(\alpha)$ is the unique solution of ..." (you don't need the "deterministic function") Furthermore, is $f$ Lipschitz continous? If not, the ODE (8) is not guaranteed to have a unique solution. The approximation in Eq. (10) is for SCM. Does this provide any insights into more general settings? I believe this situation is unrealistic in general NN, for instance the results of Sec. 3 points out to the complete opposite when training both layers. Line 149: It is being mentioned that the ODE is going to be integrated numerically. So how this gives analytical predictions? Minor comments: Please proof read the paper (there were quite some english mistakes and poorly formed phrases). For instance: Line 16: I find "behind" a little weird here. Maybe besides? Line 36: too -> to Line 39: oft-used -> often used Line 46: particular -> particularly Line 48: a a -> a Line 72: coupled ODE -> coupled ODEs Line 82: this last phrase seems quite weird to me. There are solutions of SGD dynamics that SGD does not find?? I don't know what is implied here. If a dynamical system has a set of admissable solutions, how it does not "find" it? Eqn. (1): Write the sum correctly: \sum_{m=1}^M Lines 118-119: I don't understand the "overlap" ... measure of the overlap Line 123: "we give write full expression"? Line: 124: What is a closed set of ODEs? Do you mean a set of coupled ODEs with closed form solutions? Line 147: "... SGD finds solutions with drastically different performance..." ========= post rebuttal. I read the author's response and I keep my score. I think it's a good paper.

Reviewer 2



This paper studies the learning dynamics of two-layer neural networks in the teacher-student scenario under the assumptions that the input is i.i.d. from zero-mean unit-variance Gaussian and its dimension is very large. The dynamics is set to be the online algorithm or the stochastic gradient descent (SGD) with mini-batch of single sample, and the dataset size is also assumed to be sufficiently large so that the parameters have no correlation with forthcoming samples. Thanks to these assumptions, the dynamics is governed only by the covariances of connections of the student and teacher, and the closed-form macroscopic dynamics of those covariances can be derived from the SGD dynamics itself. Using this macroscopic dynamics, the generalization error which is also characterized by the covariances only, can be accurately calculated. In the soft committee machine (SCM) setting (i.e. weight from the hidden layer to the output is constant), when only the first layer of the student is leant, it is shown that the overparameterization (the student has more nodes in the hidden layer than the teacher) generally degrades the generalization ability irrespectively of the choice of activation functions (here sigmoid (erf), linear, and ReLu are treated). Meanwhile when both layers of the student are leant, the generalization ability strongly depends on the choice of the activation function: For the sigmoid activation, the generalization error decreases'' as the overparameterization level increases'' while for the other activations the generalization error almost stays constant with respect to it. A further interesting observation is that there exist another fixed point of SGD exhibiting better generalization but cannot be found from random initialization for the linear and ReLu activations. These observations reveal the complicated interplay among the activation function, the network structure, and the algorithm, which largely affects the generalization ability. This study is thus important because it reveals the need for a more careful consideration about the generalization and the learning dynamics. Originality: Analysis of two-layer neural networks is recently drawing much attentions of the researchers in the community, but this type of analysis has never been done and the problem setting is interesting. The novelty is clear. Quality: The submission is technically sound and the theoretical prediction is well supported by experiments. The quality is high enough. Clarity: The paper is well organized and equipped with nice appendices well summarizing the detailed computations and experiments, though I think the introduction is slightly redundant. No need of large modifications. I only give the list of typos and uneasy-to-understand expressions which I found: Line 36: they have too -> they have to Line 48: a a surge -> a surge Line 56: kernel regime -> kernel regimes Line 94: (x^{\mu},y^{\mu}_{B}). What the subscript B? Line 609: (S34ff) -> (S34) Equations S26-S30: These expressions are for sigmoid activation, but it is not explicitly mentioned in Appendix B. This point should be addressed. Significance: I think this work is an important one because it provides an insight about the learning dynamics and the generalization. Explicit demonstration of the superiority of the overparameterization in the generalization error in the sigmoid activation is interesting and suggesting. Additional comments after feedback: Given the author feedback, I am impressed with the result for larger K and the difference from the mean-field result in the infinite width limit, and accordingly I increased the score. I think the fact that the authors found the different convergent point of SGD from that oft the mean-field theory is very interesting and should be stressed in the Discussion section more in the revised manuscript.

Reviewer 3



--- Added after feedback --- All reviews are in agreement that this is a strong contribution and given that consensus and the convincing nature of the author response, which carefully addressed all of my concerns, I have increased my score to a 9. I firmly believe that the technical developments in this paper will enable further research on the detailed dynamics of neural networks within and beyond the student-teacher paradigm. --- Original review ---- In this compelling and well-written paper, the author(s) derive a closed-form and tractable system of ODEs to study the dynamical evolution of neural networks using a "student-teacher" paradigm. While the student-teacher set-up, in which one attempts to learn a neural network with a neural network of similar architecture, is perhaps not the most practically significant case, this framework also for a precise formulation of notions that are often left vague, most notably "over-parameterization". Typically, a less meaningful comparison between the cardinality of the data set and the number of parameters in a neural network is used. In this case, it is much more straightforward, simply being the discrepancy between the number of student vs teacher hidden units. I found the analysis to be satisfyingly complete, deriving asymptotic expressions for sigmoidal, linear, and ReLU networks. For the soft committee machines, numerical experiments seem to agree extremely well with the asymptotic expressions, to the extent I left wondering what magnitude of output noise is required in order for the asymptotic expressions to fail. Perhaps my only major complaint about the paper is that it does not describe the numerical experiments in detail. There is a promise of eventual github links, but I would have liked to know, at least at a high level in the appendix, how they were set-up, how long they were run, what was used as a stopping criterion, etc. The paper contrasts the case of the soft committee machine in which the weight of every neuron is unity with the more general and more typical set-up of training both the pre-activation and the post-activation parameters. The results are different in the two scenarios---to a surprising degree. Some of the work on mean-field neural networks that the authors cite is related to this phenomenon, I believe: the dynamical accessibility of optimal solutions hinges on the evolution of the linear coefficients of the neurons. Regarding clarity and quality: While I have a number of minor comments on some of the logical steps in the proof, as well as the more significant point about the lack of information with regard to the numerics, on the whole the paper is sufficiently clear. Overall, I found the paper to be of a high caliber, both timely and complete. I have a few comments about the proof of the paper's main theorem: - The meaning of $\mathbb{E}_{\mu} $ is not clearly stated anywhere that I could find, but I took it mean an expectation over the initial conditions taken at time-step $\mu$ - The use of $m$ and $q$ is fairly confusing---$q$ only contains the time-dependent order parameters that involve the student, whereas $m$ contains both the student and teacher order parameters. Is it necessary to separate the two rather than just using $m$ throughout? - The step going from S11 to S12 is not well-explained. While the assumption that the order parameters are uncorrelated at $\mu=0$ seems perfectly fine, this does not necessarily imply $q^0$ is independent of $f_q(m^0, x^0)$. As a result, there appears to be an extra expectation in the first line of S11. If as written the equation is correct, I think a more detailed explanation is merited. - The discussion of the coupling trick glosses over the functions $d$ and $g$ and what properties they require *Specific comments on originality* I would not say that the question being investigated is particularly original, nor is the basic framework of using order parameters and deriving ODEs for the evolution of these parameters. As the authors acknowledge, this basic approach has existed in the statistical mechanics literature for many years. However, I would emphasize that using this approach to resolve questions regarding implicit regularization in SGD and quantify over-parameterization directly is a novelty. Below are some entirely inconsequential typos that I noticed which the author(s) may want to correct: Line 156: "case studied most commonly so far" seems like a hard thing to know Line 196: misplaced "*" Line 255: "fulfil" -> fulfill Line 285: "exemplary" is a bit awkward due to the connotation of "exceptional" rather than "typical" Fig S2 is highly pixelated Fig S5 caption "sigma" -> "\sigma" SI 548: "namely namely" SI 640: "to first order \sigma^2"

[Author Response · NeurIPS 2019]

We thank all the reviewers for their diligent reading of our paper, we address their comments in order of appearance:

**Reviewer #1** **Presentation of the theorem** We thank the reviewer for the suggestions concerning the presentation of the theorem, which we will implement in the final version. **Lipschitz** The function $f(m)$ is Lipschitz-continuous and we take our initial conditions to be deterministic by assumption A3, so the uniqueness of the solution is guaranteed. **Implications of Eq. (10)** This result only applies to networks where the first layer is trained, and we report different behaviours when we train both layers in Sec. 3. We used SGD for training in both cases, so the (indirect) implications of Eq. (10) and (15) are that firstly, the (implicit) regularisation observed when training both layers is not a property of just the algorithm, and thus secondly, for general neural networks, we still have to find the precise mechanism allowing bigger networks to generalise better. **Analytical results** We cannot solve the ODE in closed form. We obtain our analytic results by linearising the equations in the limit of small noise $\sigma$ around the fixed point at zero noise. **Line 82** We mean that there exist weights which are fixed points of SGD dynamics, in the sense that the generalisation error will stay stationary at a value $\hat{\epsilon}_g$ if SGD is started with these weights. However, starting from random initialisation, SGD finds weights with a generalisation error that is higher than $\hat{\epsilon}_g$. **Line 118** The teacher-student overlaps $R^\mu = [R_{in}^\mu]$ capture the *similarity* between the weights of the $i$th student node and the $n$th teacher node. **Line 124** By "a closed set of ODEs", we mean a set of coupled ODEs where the variables that appear in them are governed by an ODE in that set. Our ODEs do not have a known closed-form solution. **Line 147** We mean that running SGD will yield networks where the generalisation error (="performance") improves or worsens with over-parameterisation (="drastically different").

**Reviewer #2** **Message for practitioners:** We have to be careful with the widely cited claim that bigger networks are better. It is not always true and we still have to understand the range of the cases in which it is. **Eqs. S26-S30** The reviewer is right that Eqs. S26-S30 are only valid for sigmoidal activation; we will emphasise this in the revised manuscript. **Additional experiments and relation to mean field limit**: We numerically checked the behaviour of $\epsilon_g$ when the number of student hidden nodes becomes much larger, see Fig. 1. This is the same plot as Fig. 4 of the main paper, only that we extended the range of $K$ to 1000. We plot the final generalisation error after convergence. The plot demonstrates that the trend observed in the paper persists: the normalised network, where we train only the first layer and divide the network output by the number of hidden units, beats the performance of a two-layer network where we train both layers. We think our results connect with the cited mean-field analyses in that they suggest that the "distributional" fixed points corresponding to the mean-field analysis persist even down to relatively small sizes of the hidden layer. However, we also found other fixed points which are not captured by the mean field analysis, such as the one leading to the increasing generalisation error. Elaborating this connection and pinning down the differences more precisely is a very interesting direction for future research.

**Reviewer #3** **Validity of the expansion** We found the results of our expansion in good agreement with numerics up to a noise with $\sigma \simeq 0.3$. **Details on the numerical experiments:** We will collect all the parameters needed to reproduce the experiments, including the reference to codes, in one section of the appendix, instead of scattering them in various places of the text. The stopping criterion is a fixed number of steps chosen in each experiment manually to be large enough to reach a stationary point of the generalisation error, usually on the order of $10^6 N$. **Context of the MNIST experiments:** We intended to verify the qualitative validity of our result in Eq. (10) for the final test error of networks where only the first layer is trained, in a setting which violates two key assumptions of our theoretical treatment: (1) that all inputs are i.i.d. draws from the multi-normal distribution and (2) that at every step, we use a previously unseen sample $(x^\mu, y^\mu)$. Crucially, it is still another teacher that generates the labels for the images. Indeed, the plots in Fig. S7 show that

Figure 1: Asymptotic generalisation error of linear networks as a function of the number of student nodes $K$. Parameters: $N = 100, M = 4, v^* = 4, \eta = 0.01, \sigma = 0.01$.

substituting MNIST inputs (orange curve) for Gausssian inputs (blue) does not change the shape of the $\epsilon_g^* \sim L$ curve, with the minimal final test error occurring for $K = M$. **Proof precisions.** Regarding the reviewer's comments on the proof (we will clarify all points in the final version): (i) The expectation $\mathbb{E}_\mu$ is the conditional expectation conditioned on the state of the Markov chain at step $\mu$, $m^\mu$. (ii) We use $q$ for any but only *one* of the time-dependent order parameters and $m$ to denote the set of all order parameters, resp. (iii) We were indeed not precise about the step from S11 to S12; we crucially also used assumption (A3) by which the initial macroscopic state is deterministic and therefore the average $\mathbb{E}$ in that line is just an average over the first sample shown during training. (iv) We kept the discussion of the coupling trick more compact than the other parts of the proof because it is not original, but due to Wang et al., Ref. 46. We will expand this section in the final version to make the paper more self-contained. The stochastic process $b^\mu$ lives in the same space as $m^\mu$, and similarly for the deterministic process $d^\mu$. They do not require additional assumptions.

[Meta-Review · NeurIPS 2019]

This paper derives a coupled system of ODEs modelling this teacher-student setup. The authors provide an asymptotic analysis of the dynamics when only the first layer is trained, and generalization error increases with the size of the student network, and results when both layers are trained are also obtained. All reviewers agree that it is a good contribution.